# Short-Chain Fructooligosaccharide Synthesis from Sugarcane Syrup with Commercial Enzyme Preparations and Some Physical and Antioxidation Properties of the Syrup and Syrup Powder

**DOI:** 10.3390/foods12152895

**Published:** 2023-07-29

**Authors:** Sudthida Kamchonemenukool, Warathep Buasum, Monthana Weerawatanakorn, Tipawan Thongsook

**Affiliations:** Department of Agro-Industry, Faculty of Agriculture Natural Resources and Environment, Naresuan University, Phitsanulok 65000, Thailand

**Keywords:** fructooligosaccharide, FOS, sugarcane syrup, commercial enzyme preparations, syrup powder, antioxidants

## Abstract

Short-chain fructooligosaccharides (sc-FOS) are prebiotics beneficial to human health, which can be synthesized from raw material containing a high sucrose content. Sugarcane syrup (SS) without molasses removal contains sucrose as a major sugar and is a rich source of several bioactive compounds. The aim of this study is to investigate factors affecting sc-FOS synthesis from SS using commercial enzyme preparations containing fructosyltransferase activity as biocatalysts. sc-FOS content increased significantly as the sucrose concentration of SS in the reaction mixture increased up to 40% (*w*/*v*). Changes in carbohydrate compositions during the transfructosylating reaction of a pure sucrose solution and SS prepared from the two sugarcane cultivars Khon Kaen 3 and Suphanburi 50, catalyzed by Pectinex Ultra SP-L and Viscozyme L, showed similar profiles. Both enzymes showed a high ability to transfer fructosyl moieties to produce sc-FOS and a plateau of the total sc-FOS concentration was observed after 4 h of reaction time. For the pure sucrose solution and SS (Suphanburi 50), Viscozyme showed a superior ability to convert sucrose to Pectinex, with a higher sc-FOS yield (g FOS/100 g of initial sucrose), GF2 or 1-kestose yield (g GF2/g of initial sucrose) and GF3 or nystose yield (g GF3/g of initial sucrose). sc-FOS syrup (FOS SS) and the foam-mat-dried syrup powder prepared from SS and FOS SS, respectively, contained a high total phenolic content and possessed higher antioxidant activities than those prepared from pure sucrose, but contained lower calories.

## 1. Introduction

Short-chain fructooligosaccharides (sc-FOS) are molecules containing about two or four β (2–1)-linked fructosyl units with a terminal α-D-glucose residue as a mixture of 1-kestose (GF2), nystose (GF3) and 1F-fructofuranosylnystose (GF4). sc-FOS are widely used as prebiotics because they improve gastrointestinal conditions by stimulating the growth of bifidobacteria in the large intestine and are beneficial to the health of the host [1,2]. sc-FOS are synthesized from sucrose by microbial β-fructofuranosidase (FFase, EC 3.2.1.26) or fructosyltransferase (FTase, EC 2.4.1.9) under specific conditions. FTase has two enzyme activities including transfructosylating activity (Ut) and hydrolyzing activity (Uh) [3]. Commercial food-grade enzyme preparations such as Pectinex Ultra SP-L, Rohapect CM, Viscozyme L and Pectinex Smash have FTase or FFase activity and serve as a source of food-grade FOS-forming enzymes [4]. Commercial food-grade enzyme preparations with a low price, versatility and stable enzymatic activity are preferred to produce sc-FOS.

Viscozyme L and Pectinex Ultra SP-L have a high Ut to Uh ratio, notable thermal and pH stability and remarkable activity toward sc-FOS production. Therefore, they are often applied for sc-FOS production, as reported in several studies [4,5,6,7]. Viscozyme L and Pectinex Ultra SP-L are two commercial enzymes prepared from *Aspergillus aculeatus* and marketed for use in the processing of fruit juice. Pectinex Ultra SP-L is marketed as polygalacturonase that hydrolyzes (1 → 4)-d-galactosiduronic linkages in pectate and other galacturonans, while Viscozyme L is marketed as endoglucanase that hydrolyzes (1 → 3)- or (1 → 4)- linkages in -d-glucans [6]. Recent studies have shown that Viscozyme L has more FFase activity than Pectinex Ultra SP-L [4,8,9,10].

Raw materials with a high sucrose content are used to produce sc-FOS and sugarcane syrup (SS) is one of the most promising, providing a high sc-FOS yield. Hajar-Azhari et al. [5] produced sc-FOS from concentrated SS using a commercial enzyme and pH-stat bioreactor. They also discovered that the Bifidobacterium utilized the enzymatically synthesized sc-FOS to generate short-chain fatty acids as metabolites in pH-controlled conditions; thus, sc-FOS from SS show potential as a prebiotic ingredient for foods and health drink products. 

SS is prepared from many varieties of sugarcane using different methods and additives. Both sugarcane varieties and SS preparation methods influence sc-FOS production, while commercial enzymes are also utilized. This study examined the profiles of sc-FOS synthesized from SS without the removal of molasses from the two varieties of sugarcane most popularly cultivated in Thailand (Khon Kaen 3 and Suphanburi 50) using two commercial enzymes, Viscozyme L and Pectinex Ultra SP-L. Even though there was a study on sc-FOS produced from SS using a commercial enzyme, influences of sugarcane cultivars on sc-FOS production have never been studied and there has been no report on the comparison of the effectiveness of the commercial enzymes used as catalysts.

SS produced without molasses removal contains nutrients and phytochemicals inherently found in sugarcane. SS is, therefore, a rich source of several bioactive compounds including natural antioxidants. Consequently, sc-FOS syrup (FOS SS), produced from SS, not only contains beneficial prebiotic compounds but also is rich in bioactive compounds and is considered as a healthy-choice syrup. In this study, some physical properties, phenolic content and antioxidant activities of FOS SS were also determined. Moreover, FOS SS was also further processed to powder with the foam-mat drying method and its properties were investigated.

## 2. Materials and Methods

### 2.1. Enzymes and FOS Standards

Pectinex Ultra SP-L and Viscozyme L from *A. aculeatus* were purchased from Sigma-Aldrich (St. Louis, MO, USA). sc-FOS standards (1-kestose, nystose and 1F-fructofuranosylnystose) were obtained from Megazyme (Lansing, MI, USA). 

### 2.2. Sugarcane Syrup Preparation

Whole stalks of the two sugarcane varieties Khon Kaen 3 and Suphanburi 50 were harvested from Rai-Hong-Kit-Jarean Organic Farm (Bantan, Chonnabot, Khon Kaen, Thailand) as the source material. The stalks were collected at the stage of maturity (8–9 months of cultivation). The outer rinds were manually cleaned and peeled and the juice was extracted using a two-roller power crusher. Total soluble solids (TSS) of the fresh juice were measured using a refractometer (Nippon Optical Works, 507-III). One liter of juice was then filtered through a muslin cloth and transferred to an open household pan to boil at 100–120 °C. The boiling concentration process was monitored until SS at 68–75 °Brix was obtained. The SS was kept at 4 °C until further use. 

### 2.3. Determination of Sugar Compositions

Sugar concentrations in the solution were determined with high-performance liquid chromatography (HPLC). The concentration of each sample was adjusted using deionized water and filtered through a filter (0.45 μm) before an analysis. Conditions for the HPLC analyses were as follows: column, ZORBAX NH_2_ (5 µm, 4.6 mm × 150 mm; Agilent, Santa Clara, CA, USA); mobile phase acetonitrile/water (70:30 (*v*/*v*)); flow rate, 1.2 mL min^−1^; temperature, 30 ℃; and refractive index detector 5450 (Hitachi High-Tech Science, Tokyo, Japan).

Sugar concentration was determined from a standard curve plotted using the known concentrations of each sugar. Figure 1 shows a chromatogram of sugar compositions analyzed with HPLC.

### 2.4. Enzyme Assays

The reaction mixture for determining enzyme activities consisted of 40% (*w*/*v*) sucrose as the substrate prepared in a 0.1 M acetate buffer (pH 5.5) and enzyme solution (1% *v*/*v*). The total volume of the assay was 10 mL. The enzyme reaction was carried out at 50 °C for 1 h with moderate shaking in Erlenmeyer flasks and terminated by heating in boiling water. Quantitative analyses of fructooligosaccharides as kestose (Glc-Fru-Fru or GF2), nystose (Glc-Fru-Fru-Fru or GF3) and fructofuranosyl-nystose (Glc-Fru-Fru-Fru-Fru or GF4), fructose and sucrose in the reaction mixture were performed with HPLC.

Transfructosylating activity (Ut) and hydrolyzing activity (Uh) were determined by measuring the concentrations of 1-kestose and fructose with HPLC, respectively. One unit (1U) of transfructosylating activity was defined as the amount of enzyme required to transfer 1 µmol of fructose per minute. A total of 1 U of hydrolyzing activity was defined as the amount of enzyme required to release 1 µmol of free fructose per minute. A total of 1 U of transfructosylating activity was defined as the amount of enzyme required to release 1 µmol of 1-kestose per minute. FFase activity was determined from the amount of glucose produced in the reaction. A total of 1 U of the FFase activity was defined as the amount of enzyme that released 1 µmol of glucose per minute under the above assay conditions [11].

### 2.5. Production of sc-FOS from Sugarcane Syrup Using Two Commercial Enzymes 

Each sugarcane syrup was diluted to obtain sucrose at 400 ± 10 mg/mL or approximately 40% *w*/*v* (g sucrose/100 mL) using a 0.1 M acetate buffer (pH 5.5) before enzymatic treatment. An aliquot of 10 mL of the syrup sample was weighed into a flask and then treated with 3% *v*/*v* Viscozyme L or Pectinex Ultra SP-L at 55 °C for a treatment time of 0, 0.5, 1, 2, 3, 4, 5 and 6 h. The treated sugarcane syrup samples were then immediately heated in boiling water for 3 min to stop the reaction. The juice was filtered through a filter (0.45 μm) before the HPLC analysis. 

### 2.6. Preparation of FOS SS Powder with Foam-Mat Drying

SS diluted to obtain 40% *w*/*v* sucrose and FOS syrup prepared from SS at 40% sucrose (FOS SS) were transferred to powder using the foam-mat drying technique. A foaming agent (Methocel™) was incorporated into the syrup at 2% (*w*/*v*) of the syrup. Because of the high sugar content in the syrup, gum arabic was added in the formulation as a drying aid agent at 20% (*w*/*v*) of the syrup. The mixtures of the syrup, gum arabic and foaming agent were mixed in a KitchenAid mixer at a low speed for 1 min to facilitate the even distribution of the foam stabilizing agent within the mixture. Further whipping was carried out at the maximum speed for 8 min to form stiff foams. The foams were then air dried at 60 °C in a tray dryer. Moisture content was monitored at 15 min intervals until the moisture content was below 10%. Then, all dried samples were ground and sieved (100 mesh) to obtain fine powder. The powder was packed into aluminum foil bags and stored at a temperature of under 4 °C for a further analysis.

### 2.7. Characterization of SS, sc-FOS Syrup and the Syrup Powder

#### 2.7.1. Color

The ICUMSA color was determined with the preliminary dilution of samples to 1.25 °Brix (i.e., percentage of total soluble solids in the sugarcane juice). The pH was corrected to 7.0 ± 0.05. The spectrophotometric analysis was carried out at 420 nm using a UV-visible spectrophotometer (Thermo Fisher Scientific, Waltham, MA, USA) and the ICUMSA color was expressed using
$$\mathrm{Icumsa}\;\mathrm{color}\;(420\;\mathrm{nm})=\left[\frac{ABS\times 1000}{\left(density\times \frac{Brix}{100}\right)}\right]$$
$$\mathrm{Density}=1+\left(\left(Brix0\times \frac{200+BrixC}{5400}\right)\times \frac{BrixC}{Brix0}\right)$$
where ABS is the sample absorbance read at 420 nm, Brix is the total soluble solids of the diluted sample (=1.25, if the value is slightly different, it must be indicated) of sugarcane juice, Brix0 is the reading of soluble solids in the original sample and BrixC is the reading of soluble solids in the diluted sample with pH adjusted to 7.0 ± 0.05 [12].

#### 2.7.2. Viscosity

A programmable Brookfield Viscometer (Model pro DV2TRVTJ0, USA) with a spindle SC4-21 was used to determine the rheological properties of the sugarcane syrup at 25 °C at 200 rpm. Rheocalc software(version 1.2.19) produced by Brookfield Company was used to process the viscosity information using data in an Excel format. After recording values of shear stress, shear rate and viscosity, the results were analyzed using the rheological models [13].

#### 2.7.3. Evaluation of Antioxidant Compounds

##### Total Phenolic Content (TPC) Determination

Total phenolic content (TPC) was determined using the Folin–Ciocalteu method following Iqbal et al. [14] and Eggleston et al. [15] with some modifications. Gallic acid was used as a standard and results were expressed as mg gallic acid equivalent (GAE)/100 mL. Briefly, 0.3 mL of the sample was added to 1.5 mL of a 10% (*w*/*v*) Folin–Ciocalteu reagent and mixed with 1.5 mL of 7.5% (*w*/*v*) sodium carbonate. The flasks were shaken and allowed to stand for 45 min at room temperature before absorbance was read at 765 nm with a spectrophotometer (Thermo Fisher Scientific, Waltham, MA, USA). Results were recorded as the mean of three repetitive procedures, with TPC expressed as mg gallic acid equivalent (mg GAE) per 100 mL of the extract.

##### Radical-Scavenging-Activity-DPPH-Assay and ABTS Assay (EC50)

The 2,2-diphenyl-1-picryl-hydrazylhydrate (DPPH) free radical scavenging activity was measured according to the method of Boue et al. [16] with some modifications. Briefly, a 500 μL sample was allowed to react with 2 mL of a DPPH solution (0.1 mM). This mixture was then incubated at room temperature. After 30 min, the absorbance was measured at 517 nm using a spectrophotometer and then converted to the percent inhibition of the DPPH radical and the (2,2’-azino-bis (3-ethyl-benzothiazoline-6-sulfonic acid) diammonium salt radical cation (ABTS^*+^) described by Re et al. [17] with some modifications. An aliquot of 1.9 mL of an ABTS^*+^ solution was added to 100 μL of the test samples at the required concentration. The samples were mixed thoroughly, the reaction mixtures were incubated at room temperature for 20 min and the absorbance was recorded at 734 nm. Percentage inhibitions of both the standard and the samples were calculated and concentrations of DPPH and ABTS contents in the extract were reported as mg of Trolox equivalent (TE)/g of extract.

Both assays were carried out in triplicate. Percentage inhibitions for the DPPH and ABTS assays were calculated according to the following formula:Percentage radical scavenging activity = (A_control_ − A_sample_/A_control_) × 100
where A_sample_ is the absorbance of the sample and A_control_ is the absorbance of the positive control (Trolox).

#### 2.7.4. Caloric Value

The caloric value (Kcal) for each syrup was calculated with the following equation: Calories = Fructose × 4.0 + Glucose × 4.0 + Sucrose × 4.0 + FOS × 1.5. This was based on the caloric values for fructose, glucose, sucrose and oligofructose as 4.0, 4.0, 4.0 and 1.5 kcal/g, respectively [18,19,20].

## 3. Results and Discussion

### 3.1. Effect of Sucrose Concentration on sc-FOS Synthesis from Sugarcane Syrup

Sugar compositions of SS prepared from two varieties of sugarcane are shown in Table 1. Results indicated that glucose and fructose contents varied for SS produced from two sugarcane cultivars; still, sucrose is the major sugar in the SS, accounting for more than 93% of the total sugar in the SS. sc-FOS is synthesized from sucrose by FTase present in commercial food-grade enzymes. Therefore, sucrose content in the reaction mixture played a crucial role in sc-FOS synthesis. To study the effect of sucrose content on sc-FOS synthesis, SS was diluted to obtain the sucrose concentration of 20%, 40% and 50% (*w*/*v*), and then used as starting raw materials for sc-FOS synthesis. Figure 2 shows that a significantly higher amount of sc-FOS was synthesized over time for SS containing 40% and 50% sucrose compared to SS containing 20% sucrose (*p* < 0.05). The sc-FOS production rate is proportional to the initial concentrations of sucrose and the enzyme. At the same enzyme concentration, as sucrose concentrations increased, the sc-FOS content in the reaction mixture also increased and this was commonly observed in other studies [21,22]. Even though SS containing 50% sucrose produced more sc-FOS, due to the limitation of the raw material, the rest of the studies were based on SS containing 40% sucrose.

### 3.2. Effect of SS Cultivars and Enzyme Sources on sc-FOS Synthesis from Sugarcane Syrup

Changes in carbohydrate compositions during the transfructosylating reaction of approximately 40% sucrose catalyzed by Pectinex Ultra SP-L compared with Viscozyme L are shown in Figure 3a,b. Similar experiments were conducted for sugarcane syrup (SS) from two sugarcane cultivars including Khon Kaen 3 (Figure 3c,d) and Suphanburi 50 (Figure 3e,f) with sucrose concentration adjusted to 40%.

Both Viscozyme L and Pectinex Ultra SP-L contained FFase and their activities are shown in Table 2. Sucrose conversion to sc-FOS by FFase started with the cleavage of sucrose resulting in glucose as a byproduct. Subsequently, fructose was transferred to sucrose (GF) to produce 1-kestose (GF2) or transferred to a saccharide acceptor such as 1-kestose (GF2) to produce nystose (GF3) or nystose (GF3) to produce 1F-fructofuranosylnystose (GF4). As the reaction progressed, an increase in glucose content was observed whereas fructose content was almost unchanged (2–3%). The low content of fructose produced indicated a high ability of the enzymes to transfer fructosyl moieties, the product of the hydrolysis reaction of sucrose, to couple to another sucrose molecule for sc-FOS production. This resulted in glucose units in the reaction medium [23,24].

As the reaction progressed, sucrose was rapidly converted into sc-FOS during the first 2 hours. The rates of conversion were lower after the second hour for all samples (Figure 3a,c,e). Total sc-FOS concentration was rather stable after 3 h of reaction time because sucrose concentration decreased gradually as the reaction time increased (Appendix A), whereas an increase in glucose during the reaction acted as a competitive inhibitor of sc-FOS synthesis [5,25,26]. It was noticed that in the case of SS from Khon Kaen 3 (Figure 3e), the total saccharide concentration at the end of the process could be higher than the total initial concentration of sucrose, fructose and glucose due to the decomposition of cellulosic or starchy components by some enzymes in the commercial enzyme preparations. These components were possibly considered as impurities during the juice extraction process, which later contaminated the SS.

At the same enzyme content (3% *v*/*v*), the profiles of total sc-FOS, glucose, fructose, GF2, GF3 and GF4 showed similar trends and concurred with previously reported results for sc-FOS synthesis from pure sucrose (500 mg/mL or 50% sucrose), while sc-FOS concentration remained almost constant after 3 h [4]. A similar trend was reported for sc-FOS synthesis from 40% *w/v P. nigra* flour suspensions containing 19.39 ± 0.17% *w/v* sucrose and 20% *w/v* sucrose using 3% *v/v* Viscozyme L. At high concentrations of sucrose, the maximum sc-FOS was reached after 3 h, which then stabilized for the remainder of the 6 h reaction time [10]. The sc-FOS production profiles produced from beet sugar syrup and molasses containing 620 and 570 mg/mL of sucrose using Pectinex Ultra SP-L [27] remained almost constant after 30 h and 65 h, respectively. The time difference to reach the maximum sc-FOS content was due to different amounts of the enzyme used. On the contrary, different profiles of sc-FOS production were observed when initial 6% SS (*v*/*v*) prepared from concentrated SS containing 58.93% *w/v* accounting for only 3.53% (35.3 mg/mL) was used as initial sucrose in the reaction system (Hajar-Azhari et al. [5]). The starting sucrose concentration of 3.53% in this study was lower than in our study (40%). This low starting sucrose concentration resulted in different sugar profiles compared to our study (more than 10 times the initial sucrose concentration), while nystose (GF3) and 1F-fructosylnystose (GF4) were not observed during the 6 h reaction. 

The sc-FOS production rate is proportional to the initial concentrations of sucrose and the enzyme. As sucrose and enzyme concentrations increased, the amount of sc-FOS in the reaction mixture also increased [24,28]. The sc-FOS production profiles produced from SS reported by Hajar-Azhar et al. [5] showed a continuously increasing sc-FOS content over 6 h, as expected. Despite the fact that 4% (*v*/*v*) Viscozyme L (36 FU/mL; FU: fructosyltransferase units) was applied, the sc-FOS yield was lower than in our study because sucrose concentration started at a lower level. At this condition, as glucose was generated at a low concentration, no inhibition effect of glucose was observed, and sc-FOS concentration continuously increased over 6 h.

GF2 was the first product, with a content always higher than GF3 and GF4. This observation can be explained because, in the first steps of the sc-FOS synthesis reaction, sucrose acts as a fructosyl donor and acceptor, leading to the simultaneous production of FOS with longer chains (GF + F or GF2) and glucose and fructose [29]. The production of GF2 reached the highest after 3 h of reaction time, when total sc-FOS concentration was quite stable (Figure 3). GF2 content increased as the reaction time progressed and decreased after 4 h (Figure 3b,d,f).

GF2 depletion was observed for Viscozyme but not for Pectinex. When the decrease in GF2 was observed, the total sc-FOS content remained steady while GF3 and GF4 contents increased. This was also observed in another study by Vega-Paulino and Zuniga-Hansen [4] and they suggested that this reaction is independent of the overall kinetic control mechanism because sc-FOS were not transformed into fructose and glucose as the reaction time progressed. 

Viscozyme and Pectinex were obtained from the same microorganism; therefore, the reduction in GF2 content was observed over an extended period. Depletion in GF2 content was observed in the sc-FOS production profiles of beet sugar syrup and molasses using Pectinex Ultra SP-L but this occurred after 20 h of the reaction [4,27].

The time course of the reaction (Figure 3b,d,f) indicated that saccharides showing a higher degree of polymerization (GF3 and GF4) increased with the progress of the reaction, although the total amount of oligosaccharides did not greatly change after 3–4 h. As the rate of GF2 production reduced, GF3 and GF4 were constantly generated, keeping the total sc-FOS steady. The sc-FOS molecule occurrence and the ratio of individual fractions (GF2, GF3 and GF4) constantly changed. At the beginning of the reaction, GF2 represented the major component of the system, while at longer time intervals, the ratio of GF3 or GF4 to GF2 increased [7].

A similar change in sugar profiles was observed during sc-FOS production from SS containing 40% sucrose regardless of the variety of the sugarcane. The profiles were not unlike those of pure sucrose at the same concentration. However, changes in the yield of sc-FOS GF2 and GF3, as amounts of sc-FOS, GF2 and GF3 produced over time compared with the initial sucrose content in the reaction mixture during the transfructosylating reaction, revealed influences of the enzyme sources. 

Despite similar changes in sugar profiles, Viscozyme showed an ability to convert sucrose in the sucrose solution and Suphanburi 50 syrup to sc-FOS superior to Pectinex (Figure 4a,b). Similarly, GF3, GF4 and glucose contents were also higher for Viscozyme (Figure 3). These results concurred with the higher transfructosylating activity (Ut) of Viscozyme (Table 2). In agreement with our study, Vega-Paulino and Zúniga-Hansen [4] showed that sc-FOS produced per ml of enzyme (R value: g sc-FOS/mL of enzyme preparation) of Pectinex Ultra SP-L was 15.9 ± 0.1, and lower than Viscozyme L (65.7 ± 1.3). This result was obtained for 536.2 g/L of sucrose in a 50 mM sodium acetate buffer (pH 5.5) and 9 UT/g of sucrose at 50 °C and pH 5.5 for 6 h.

The FOS yield (g sc-FOS/100 g of initial sucrose) for Viscozyme using pure sucrose as a substrate was 58.8 ± 1.2 [4] and lower than in our study (71.55 ± 1.40) likely due to different enzyme contents. Lorenzoni et al. [5] reported an sc-FOS yield at 55% for the immobilized partial purification of fructofuranosidase from Viscozyme L. 

For Pectinex, the sc-FOS yield was reported at 60.44% [28] for initial sucrose at 450 g/L and 61.43% for initial sucrose at 450 g/L [27] compared to our study at 63.21 ± 0.22%. However, the time to reach the maximum yield varied with different conditions applied for each study, as well as diverse substrate and enzyme concentrations. 

In the case of Khon Kaen 3 syrup, the sc-FOS yield at 4 and 5 h showed no difference between Viscozyme and Pectinex (Figure 4c). Unlike the sc-FOS yield, the GF2 yield showed a different pattern. For Viscozyme, the GF2 yield decreased after 3 h of the reaction and the contents were either not different or lower than Pectinex (Figure 4d–f). In all cases, GF3 yields were higher for Viscozyme compared with Pectinex (Figure 4g–i), indicating that the hydrolysis reaction of GF2 occurred when the sc-FOS yield reached 60–65% for Viscozyme. However, this was not the case for Pectinex, where GF2 remained stable after 4 h. 

Regardless of the type of enzyme, higher sc-FOS and GF3 yields were observed when using the sucrose solution as a starting raw material (Figure 4a–c,g–i). This finding agreed with Hajar-Azhari et al. [5], who compared SS with a pure sucrose solution at an equivalent concentration of 10% (*w*/*v*) for sc-FOS synthesis. The sucrose conversion to sc-FOS in a percentage or sc-FOS yield observed in SS was lower (32.22 ± 0.45% or g sc-FOS/100 g of sucrose) than that of pure sucrose (39.55 ± 0.16%). In the same vein, when *P. nigra* flour suspensions containing 19.39 ± 0.17% *w/v* sucrose were used for sc-FOS synthesis, the sc-FOS yield obtained was 50% and lower than the sc-FOS yield of 55% obtained from 20% *w/v* sucrose solutions. The decrease that occurred in the sc-FOS yield when the pure sucrose solution was replaced by other substrates containing a high sucrose content was probably caused by an interruption in enzyme–substrate interaction caused by other components in the syrup.

The trend was different for GF2, where changes in the GF2 yield and content were similar for the sucrose solution and Khon Kaen 3 syrup (Figure 4d,f). For Khon Kaen 3 syrup, the sc-FOS yield remained stable after 4 h, with no significant differences found between both enzymes. However, the sc-FOS composition was expected to be different according to the enzyme used. A ratio of GF3 to GF2 or the longer-chain sc-FOS yield was higher for Viscozyme as the reaction extended beyond 4 h.

### 3.3. Characterization of Sugarcane and sc-FOS Syrup

SS, commercial FOS syrup and sc-FOS syrup (FOS SS) prepared from SS with different synthesis conditions were studied for their physical properties including total soluble solids (TSS), color (ICUMSA) and viscosity. Phenolic content and antioxidant activity with a DPPH and ABTS assay as well as total calories of the syrup were also determined, as shown in Table 3.

TSS of commercial FOS syrup (77 °Brix) were statistically higher than SS and FOS SS (68–70 °Brix). Corresponding to its high TSS, commercial FOS syrup showed the highest viscosity (158.60 mPa.S) (*p* < 0.05). FOS SS prepared with the starting sucrose content of 40% was slightly more viscous than FOS SS prepared with the starting sucrose content of 20% and SS (*p* < 0.05). As commercial FOS syrup was prepared from the sucrose solution, its color was light brown unlike SS and FOS SS, which exhibited a dark brown color, relevant to their high ICUMSA color values.

Antioxidant values including the total phenolic content (TPC), DPPH scavenging ability and ABTS scavenging ability of SS and FOS SS were significantly higher than those of commercial FOS syrup (*p* < 0.05). Phenolic contents and antioxidant values for scavenging free radicals with ABTS increased as SS in the reaction mixture increased.

Antioxidant properties of FOS SS were high because sugarcane juice as the raw material of SS is a rich source of natural antioxidants [30]. Table 3 shows higher levels of antioxidant activity identified through TPC (196.70 ± 5.93 mg GAE/100 mL), DPPH (7.03 ± 0.25 mg/mL) and ABTS (4.34 ± 0.01 mg/mL) than those of commercial FOS syrup (*p* < 0.05), thus confirming the remaining phenolic compounds and antioxidant potential in SS. Ali et al. [31] reported TPC and antioxidant activity (DPPH and ABTS) values of 93 ± 2.9 mg GAE/100 g of extract, 2.34 ± 0.66 TE/g of extract and ABTS at 9.65 ± 0.26 mg TE/g of extract in sugarcane juice. These results are similar to Kerdchan and Srihanam [32] who found that all sugarcane cultivars had similar TPC in the range of 115–123 mg GAE/100 g DW.

The caloric value of the syrup in Kcal per 100 mL was calculated from the total energy content of the amount of each sugar component in the syrup, as shown in Table 3. FOS SS contained lower calories than SS (*p* < 0.05), as more than half of the sucrose was converted to FOS.

### 3.4. Drying the Syrup Foam and Characteristic of FOS SS Powder

Foam-mat drying is a mild-drying technology that removes water from a foam obtained by the whipping of a liquid or semi-liquid food added with a foaming agent. Foam-mat drying is a desirable drying process suitable for a high sugar content and sticky materials. Unlike other drying methods (e.g., spray or drum drying), which usually cause problems when drying high-sugar-content materials due to a high temperature exposure, foam-mat drying uses a much lower temperature for removing water from the raw material. Besides its cost effectiveness of production, foam-mat drying holds benefit in terms of the retention of nutritional quality. Figure 5 shows drying curves of pure sucrose syrup, SS and FOS SS foam during drying at 60 °C. The result indicated that there was a similarity of the drying behavior of SS and FOS SS. The slight difference in the drying curve of the control (pure sucrose) was likely due to its hygroscopic property.

Sugar compositions, total phenolic content and antioxidation properties of the foam-mat-dried syrup powder samples are shown in Table 4. The TPC or total phenolic content (expressed as GAE on dry-mass basis) of the syrup powder ranged from 0.25 mg/g in the control (pure sucrose) to 1.77 mg/g in the SS and FOS SS. The antioxidant capacity (AOC) values measured with the DPPH assay (Table 4), expressed as TE on a dry-mass basis, ranged from 1.86 (control) to 3.39 mg/g (SS) of the syrup powder. Additionally, the AOC measured with the ABTS assay ranged from 1.86 (control) to 6.45 mg/g (SS) of the syrup powder. The TPC and AOC values were significantly lower (*p* < 0.05) for the control or syrup powder prepared from pure sucrose than those prepared from SS and FOS SS due to the contribution of SS in the syrup powder. Bioactive compounds such as phenolic acid (caffeic, chlorogenic and coumaric acids) and flavonoids (naringenin, tricin, apigenin and luteolin derivatives) were found in non-centrifugal sugarcane products and potentially contributed to the antioxidation capacities of SS and FOS SS. Results from Table 3 and Table 4 lead to the conclusion that sc-FOS synthesis from SS did not alter bioactivities of SS and both forms of sc-FOS from SS, syrup or powder maintained bioactive compounds of SS.

## 4. Conclusions

This study highlights factors affecting sc-FOS synthesized from SS. The sucrose concentration of SS significantly influenced the sc-FOS yield. Sugarcane cultivars showed no influence on the sc-FOS production profile, considering that the syrup prepared contained an equivalent sucrose concentration. Viscozyme has higher transfructosylating activity (Ut) than Pectinex Ultra SP-L, and in most cases, gave a higher sc-FOS yield (62% for Viscozyme compared to 56% for Pectinex) (*p* < 0.05). However, Pectinex Ultra SP-L can be considered as an alternative food-grade commercial enzyme for sc-FOS production from SS as the sc-FOS yield of 61–62% was obtained from SS (Suphanburi 50) for both enzymes. sc-FOS synthesis from SS did not alter bioactivities of SS and both forms of sc-FOS from SS, syrup or powder maintained a higher total phenolic content (237 compared to 36 mg GAE/100 mL of syrup and 1.77 compared to 0.25 mg GAE/g of powder, db) and greater antioxidant activities (EC50 values for ABTS of 6.04 compared to 1.91 mg/mL of syrup and 6.33 compared to 1.85 mg eq Trolox/g of powder db) with lower calories (144 compared to 289 Kcal/100 mL of syrup and 2.21 compared to 3.46 cal/g of powder, db) compared to sc-FOS syrup prepared from pure sucrose (*p* < 0.05). Therefore, sc-FOS syrup synthesized from SS could serve as a valuable health food ingredient for food and drink products.

## Figures and Tables

**Figure 1 foods-12-02895-f001:**
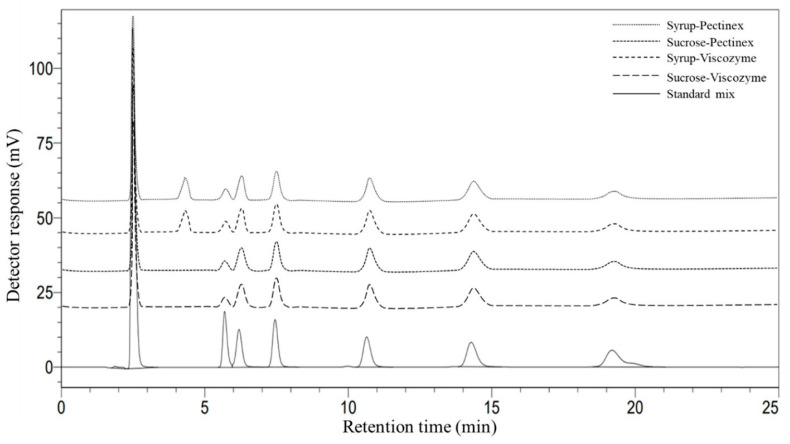
Characteristic chromatogram of the reaction mixture: fructose (tr = ~5.7 min), glucose (tr = ~6.2 min), sucrose (tr = ~7.5 min), 1-kestose-GF2 (tr = ~10.6 min), nystose-GF3 (tr = ~14.3 min) and 1F-fructofuranosylnystose-GF4 (tr = ~19.2 min), where tr is the retention time of individual components.

**Figure 2 foods-12-02895-f002:**
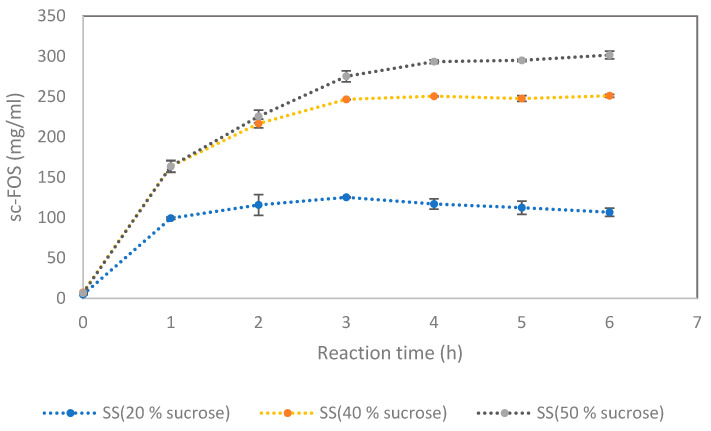
Time course of sc-FOS synthesis catalyzed by Viscozyme L in 50 mL shake flasks at 50 °C, pH 5.5 and 150 rpm. Substrate: SS (Khon Kaen 3) containing 20%, 40% and 50% sucrose (*w*/*v*). Enzyme concentration: 3% (*v*/*v*). Results correspond to the average of 2 independent assays.

**Figure 3 foods-12-02895-f003:**
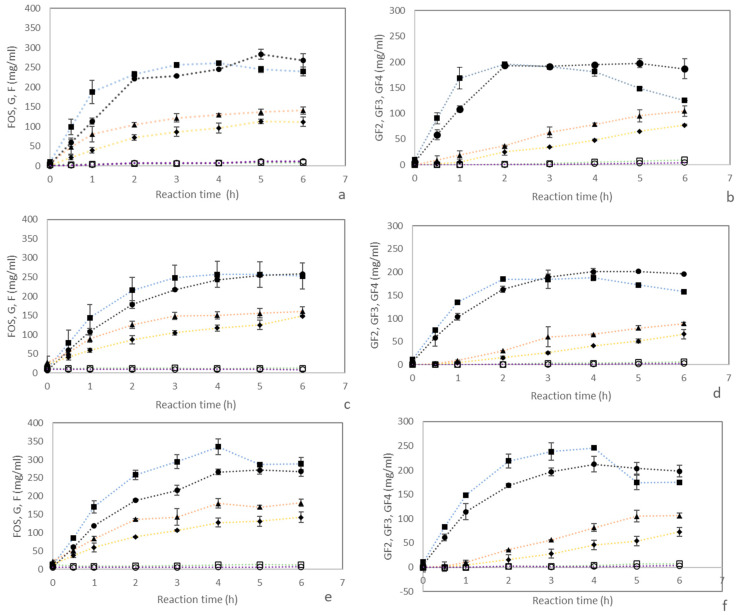
Time course of sc-FOS synthesis catalyzed by Viscozyme L and Pectinex Ultra SP-L in 50 mL shake flasks at 50 °C, pH 5.5 and 150 rpm. Substrate: 40% *w/v* sucrose (**a**,**b**), SS at 40% *w*/*v* sucrose (Suphanburi 50) (**c**,**d**), SS at 40% *w*/*v* sucrose (Khon Kaen 3) (**e**,**f**). Enzyme concentration: 3% (*v*/*v*). Results correspond to the average of 2 independent assays ± confidence interval (95% confidence level). FOS and GF2 for Viscozyme shown as blue dotted line and, for Pectinex, shown as black dotted line. Glucose and GF3 for Viscozyme shown as orange dotted line and, for Pectinex, shown as yellow dotted line. Fructose and GF4 for Viscozyme shown as green dotted line and, for Pectinex, shown as purple dotted line.

**Figure 4 foods-12-02895-f004:**
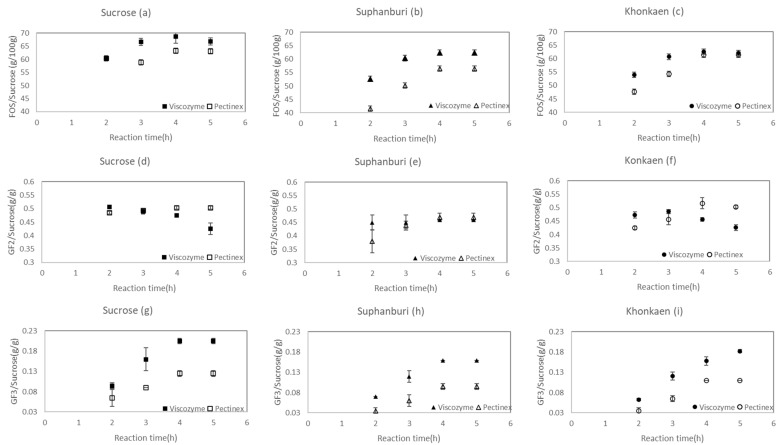
sc-FOS yield (%) or sc-FOS/sucrose (g/100 g) (**a**–**c**), GF2/sucrose (g/g) (**d**–**f**) and GF3/sucrose (g/g) (**g**–**i**) of sc-FOS synthesis catalyzed by Viscozyme L and Pectinex Ultra SP-L in 50 mL shake flasks at 50 °C, pH 5.5 and 150 rpm. Substrate: 40% *w/v* sucrose solution and SS at 40% *w*/*v* sucrose (Suphanburi 50 and Khon Kaen 3). Enzyme concentration: 3% (*v*/*v*).

**Figure 5 foods-12-02895-f005:**
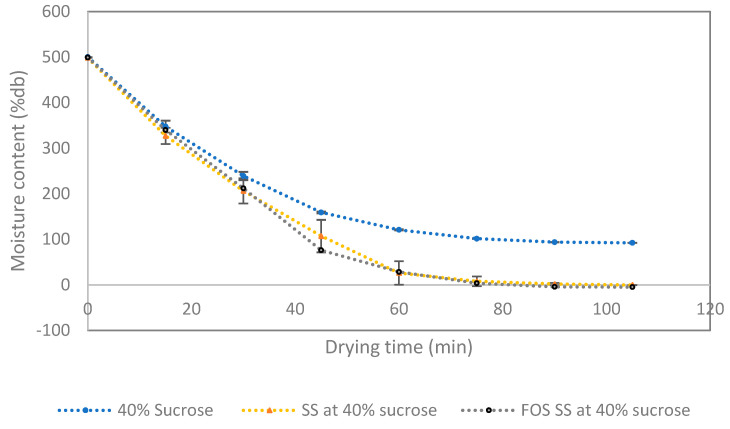
Changes in moisture content of syrup foam dried at 60 °C. The foam prepared from pure sucrose syrup, SS and FOS SS containing 40% sucrose.

**Table 1 foods-12-02895-t001:** Sugar contents of sugarcane syrup.

Sugarcane Variety	Sugar Content (mg/mL)		
	Fructose	Glucose	Sucrose
Khon Kaen 3	11.32 ± 0.20 ^b^	13.81 ± 0.24 ^b^	759.14 ± 0.36
Suphanburi 50	20.75 ± 0.42 ^a^	28.51 ± 0.18 ^a^	755.36 ± 0.53

Each value represents the mean ± S.D. Values with different superscript letters in the same column are significantly different (*p* < 0.05).

**Table 2 foods-12-02895-t002:** Enzyme activity of Viscozyme L and Pectinex Ultra SP-L.

Enzyme	Uf	Ut	Uh
Viscozyme L	26.89 ± 1.509	20.62 ± 1.24	1.07 ± 0.19
Pectinex Ultra SP-L	14.32 ± 0.61	12.59 ± 0.43	nd

nd: not detected.

**Table 3 foods-12-02895-t003:** Characterization and antioxidant value of SS, commercial FOS syrup and FOS syrup prepared from SS.

Sample	°Brix	ICUMSA Color	Viscosity (mPa.S)	Total Phenolic (mgGAE/100 mL)	EC50 Values for DPPH Assay (mg/mL)	EC50 Values for ABTS Assay (mg/mL)	Caloric Value (Kcal/100 mL)
**SS (Khon Kaen)**	68 ± 0.12 ^c^	6676.41 ± 84.41 ^e^	128.53 ± 0.55 ^f^	196.70 ± 5.93 ^d^	7.03 ± 0.25 ^c^	4.34 ± 0.01 ^d^	313.71 ± 0.03 ^b^
**SS (Suphanburi)**	68 ± 0.11 ^c^	6505.85 ± 73.10 ^f^	131.93 ± 0.60 ^e^	179.08 ± 1.66 ^e^	6.30 ± 0.17 ^d^	4.07 ± 0.02 ^e^	321.85 ± 0.03 ^a^
**FOS SS: 20% ^#^**	70 ± 0.11 ^b^	7285.58 ± 42.20 ^d^	132.33 ± 0.55 ^e^	233.31 ± 2.92 ^c^	8.39 ± 0.05 ^b^	5.80 ± 0.06 ^c^	137.64 ± 2.92 ^e^
**FOS SS: 40% ^#^**	70 ± 0.06 ^b^	7602.34 ± 73.10 ^c^	135.83 ± 0.60 ^c^	237.64 ± 0.88 ^b^	8.66 ± 0.11 ^ab^	6.04 ± 0.10 ^b^	144.82 ± 2.20 ^e^
**Commercial FOS syrup**	77 ± 0.06 ^a^	603.86 ± 41.84 ^g^	158.60 ± 0.46 ^a^	36.32 ± 0.76 ^f^	0.41 ± 0.01 ^e^	1.91 ± 0.05 ^f^	289.52 ± 7.25 ^c^

Values with different lowercase superscript letters in the same column are significantly different (*p* < 0.05). ^#^ FOS SS (Khon Kaen) was prepared using SS with starting sucrose content of either 20 or 40% with 3% Pectinex Ultra SP-L at 55 °C. Total reaction time was 5 h.

**Table 4 foods-12-02895-t004:** Some physical and antioxidation properties of syrup powder.

Sample	Moisture Content (%)	Sugar Composition (g/g, db)				Antioxidant Capacity		Total Phenolic (mg GAE/g db)	Calories (cal/g db)
		Sucrose (GF)	GF2	GF3	GF4	DPPH Assay (mg eq Trolox/g db)	ABTS Assay (mg eq Trolox/ g db)		
**SS**	2.00 ± 0.19 ^b^	0.80 ± 0.00	nd	nd	nd	3.39 ± 0.22 ^a^	6.45 ± 0.22 ^a^	1.76 ± 0.04 ^a^	3.40 ± 0.00 ^b^
**FOS SS**	2.24 ± 0.39 ^b^	0.11 ± 0.00	0.37 ± 0.00	0.11 ± 0.00	Trace	3.36 ± 0.12 ^a^	6.33 ± 0.17 ^a^	1.77 ± 0.03 ^a^	2.21 ± 0.01 ^c^
**Sucrose**	3.58 ± 0.25 ^a^	0.87 ± 0.00	nd	nd	nd	1.86 ± 0.10 ^b^	1.85 ± 0.18 ^b^	0.25 ± 0.02 ^b^	3.46 ± 0.00 ^a^

Values with different lowercase superscript letters in the same column are significantly different (*p* < 0.05). SS is syrup powder prepared from SS containing 40% *w*/*v* sucrose. FOS SS is FOS syrup powder prepared from SS containing 40% *w*/*v* sucrose. Sucrose is syrup powder prepared from 40% *w/v* sucrose. nd: not detected. db: dry basis.

## Data Availability

The data used to support the findings of this study can be made available by the corresponding author upon request.

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
