# Peer review of "Short-Chain Fructooligosaccharide Synthesis from Sugarcane Syrup with Commercial Enzyme Preparations and Some Physical and Antioxidation Properties of the Syrup and Syrup Powder"

_foods, 2023, doi:10.3390/foods12152895_

Round 1
Reviewer 1 Report
The manuscript entitled 'Short-chain fructooligosaccharide synthesis from sugarcane syrup by commercial enzyme preparations and some physical and antioxidation properties of the syrup and syrup powder' describes the potential synthesis of FOC prebiotics from sugarcane using 2 commercial enzymes. The FOS yield, antioxidant and other characterization has been conducted by the authors. However, the manuscript is not suitable to be accepted in its present form due to the discussion is not in-depth and the authors fail to highlight the novelty of the study. The abstract and the conclusion need to be re-write to be more comprehensive. There are also several statement especially in the Results and discussion section that requires further clarifications from the authors. Please refer to the comments in the attachment.

Some of the sentences in the manuscript requires rephrasing as the sentences are hard-to-read and might confusing the readers.
Reviewer 2 Report
This work deals with the preparation of fructooligosaccharides using two commercial enzyme preparations. This research was carried out before by other authors. A relatively recent paper of Hajar-Azhari et al. (cited ref. no. 5) used a dilute sugarcane syrup as a substrate. The declared novelty of the manuscript is that the process can be carried out in a concentrated sugarcane syrup. According to my opinion, there is no real novelty. FOS are typically produced using concentrated sucrose solutions because hydrolysis is suppressed and the costs for water removal are lower. Inhibitory effects of non-saccharide components were not observed in dilute solutions in the previous paper. It is therefore not surprising that they were not observed here either. I am not an expert in food technology, but I do not see much scientific value in the use of two different sugarcane varieties or determination of phenolics.
Moreover, I have found some inconsistencies and flaws in the presented results. The authors declare that complete sucrose conversion was after a few hours of the reaction process. This is in a complete disagreement with rich literature results. It is well known that the sucrose conversion reaches rather quickly the value of about 90 % when the total FOS concentration approaches a maximum. Further reaction progress is manifested by the increase of the degree of polymerization but without a significant increase of sucrose conversion. This was also demonstrated for the beta-fructofuranosidases contained in Pectinex Ultra SP-L and Viscozyme L. Sucrose serves as the only donor of fructosyl moieties. These enzymes catalyze only hydrolysis of FOS but not their transfructosylation.
Material balances do not match for the presented courses in Fig. 4. The sucrose initial concentration was 400 g/L. It is impossible to reach the total saccharide concentration around 500 g/L during the reaction progress as it is shown in Figs. 4e and 4f. Another impossible result is that the total FOS and glucose formation stopped but the transformation of kestose (GF2) to nystose (GF3) continued. It is essentially in all three experiments, but this inconsistency is the most visible in Fig. 4c and 4d for Viscozyme. Between the 3rd and 6th, the total FOS concentration is almost constant at 250 g/L and glucose concentration at 150 g/L, which indicates that sucrose was fully converted. But the Gf2 concentration decreased from about 190 g/L to 150 g/L and the GF3 concentration increased from about 60 g/L to 90 g/L. This transformation is however impossible without release of glucose.
Author Response
"Please see the attachment."

Round 2
Reviewer 1 Report
The manuscript entitled 'Short-chain fructooligosaccharide synthesis from sugarcane syrup by commercial enzyme preparations and some physical and antioxidation properties of the syrup and syrup powder' has been significantly improved by the authors and is in much better presentation than before. Hence, the manuscript is acceptable for publication.
Author Response
The reviewer has not asked for more adjustments, therefore I did not make anymore modifications on the manuscript.
Reviewer 2 Report
My main objection to the quality of this work was that there is no real novelty in carrying out fructooligosaccharide production using a concentrated sugarcane syrup. The authors in their response did not dispute this argument. They instead underscored that they determined their antioxidant properties. As I wrote in my original review, I do not see much scientific merit in these measurements. But, I am not an expert in the area of food technology so if the editor has a different opinion about the scientific quality of the paper, I shall respect it.
Several other comments of mine were not properly addressed either.
1. They did not understand my comment about the complete conversion of sucrose. They made a wrong correction but the problem remained. It is mentioned in Line 273 that sucrose was exhausted. As I argued before it is not possible because transfructosylation would stop.
2. I could accept their explanation that the total saccharide concentration at the end of the process could be higher than the total initial concentration of sucrose, fructose and glucose due to the decomposition of cellulosic or starchy components. This explanation should also be incorporated into the manuscript.
3. The authors did not answer my final objection but they argued that the same results were achieved by other authors. I must say that they are wrong. The copied figure from that paper shows clearly that sucrose was not exhausted but rather decreased gradually until the end of the experiment. Similarly, glucose increased during the whole experiment.
